# Statistical Optimization of Flavonoid and Antioxidant Recovery from Macerated Chinese and Malaysian Lotus Root (*Nelumbo nucifera*) Using Response Surface Methodology

**DOI:** 10.3390/molecules26072014

**Published:** 2021-04-01

**Authors:** Sze-Jack Tan, Chee-Keong Lee, Chee-Yuen Gan, Olusegun Abayomi Olalere

**Affiliations:** 1Main Campus, Tunku Abdul Rahman University College, Jalan Genting Klang, Kuala Lumpur 53300, Malaysia; tansj@tarc.edu.my; 2Bioprocess Division, School of Industrial Technology, Universiti Sains Malaysia, Gelugor 11800, Malaysia; cklee@usm.my; 3Analytical Biochemistry Research Center (ABrC), sains@usm Campus, Universiti Sains Malaysia, University Innovation Incubator Building, Lebuh Bukit Jambul, Bayan Lepas 11900, Malaysia

**Keywords:** antioxidant, flavonoid, lotus root, *Nelumbo nucifera*, optimization

## Abstract

In this study, the combination of parameters required for optimal extraction of anti-oxidative components from the Chinese lotus (CLR) and Malaysian lotus (MLR) roots were carefully investigated. Box–Behnken design was employed to optimize the pH (X_1_: 2–3), extraction time (X_2_: 0.5–1.5 h) and solvent-to-sample ratio (X_3_: 20–40 mL/g) to obtain a high flavonoid yield with high % DPPH_sc_ free radical scavenging and Ferric-reducing power assay (FRAP). The analysis of variance clearly showed the significant contribution of quadratic model for all responses. The optimal conditions for both Chinese lotus (CLR) and Malaysian lotus (MLR) roots were obtained as: CLR: X_1_ = 2.5; X_2_ = 0.5 h; X_3_ = 40 mL/g; MLR: X_1_ = 2.4; X_2_ = 0.5 h; X_3_ = 40 mL/g. These optimum conditions gave (a) Total flavonoid content (TFC) of 0.599 mg PCE/g sample and 0.549 mg PCE/g sample, respectively; (b) % DPPH_sc_ of 48.36% and 29.11%, respectively; (c) FRAP value of 2.07 mM FeSO_4_ and 1.89 mM FeSO_4_, respectively. A close agreement between predicted and experimental values was found. The result obtained succinctly revealed that the Chinese lotus exhibited higher antioxidant and total flavonoid content when compared with the Malaysia lotus root at optimum extraction condition.

## 1. Introduction

Oxidative diseases such as cancer are the leading cause of deaths, and this accounted for 7.6 million global deaths in the year 2008 alone, as reported by Cancer Research UK [1]. The future projection had forecasted 13.1 million deaths in the year 2030 [1]. According to the study conducted by Sharma et al. [2], low antioxidant status and increased levels of oxidative stress biomarkers were observed in cancer patients. This indicated that the supplementation of natural antioxidants plays an important role in altering the efficacy of cancer chemotherapy. However, synthetic antioxidants that are widely used in the food and pharmaceutical industry were suspected to be carcinogenic [3,4]. Therefore, special attention has been shifted from synthetic antioxidants to the search for natural antioxidants which is advantageous and does not carry any negative effects on human health [5]. Flavonoid is a large class of secondary metabolites produced by higher plants that are challenged with environment stresses. Besides the effects as antioxidants and free radical scavengers, flavonoids also exert other biological effects such as antibacterial, antidiabetic, antiviral and anti-inflammatory actions [6,7].

Lotus plant is widely distributed throughout East Asia, China, Australia and India, not only for holding religious significance but also well known for its nutritional and medicinal value [8]. Various studies were conducted on the isolation of natural flavonoids from lotus (*Nelumbo nucifera*) parts including seeds, leaves, flowers and rhizomes [9,10]. All the parts of the lotus are reported to possess a significant amount of flavonoids and exhibit convincing antioxidant activities [11]. To date, there are limited studies on the determination of effective conditions in the extraction of flavonoids from lotus root. It is important to know that the lotus species grown in Malaysia was never explored. The preliminary study showed that these flavonoids were highly water-soluble and hardly extracted by organic solvents such as ethanol, methanol and acetone. The main objectives of this study were to compare the flavonoid contents and activities from both Malaysian lotus roots (MLR) and the Chinese lotus roots (CLR) and to access the effect of pH, extraction temperature, extraction time, and buffer-to-sample ratio in the extraction of flavonoids. The extraction conditions were then optimized using Response Surface Methodology (RSM) in order to determine the best combination of parameters for optimal flavonoid recovery.

## 2. Results and Discussion

### 2.1. Investigating the Effects of the Parameters on Antioxidant Activities

In this study, the effects of four parameters on the antioxidant capacity of the extracts were tested using a single-factor experiment. Parameters effects of variables such as the pH, extraction time, solvent-to-sample ratio and extraction temperature on the total flavonoid content (TFC), DPPH scavenging activity and ferric-reducing antioxidant power (FRAP) were further succinctly investigated as presented in the preceding sub-sections.

### 2.2. Effect of pH

The extraction process was conducted by varying the pH condition within the range of 2 and 6. The result obtained showed an increasing trend in the pH values which invariably triggered a corresponding decrease in the TFC yield for both Malaysia lotus (MLR) and China lotus (CLR) roots extracts, as presented in Figure 1. As the pH rises to a threshold of 2, a higher total flavonoid content was recorded for Malaysia lotus (MLR) and China lotus (CLR) roots extracts with values 0.699 mg PCE/g sample and 0.533 mg PCE/g sample, respectively. The results obtained indicated that the acidic condition is more effective in the total recovery of flavonoid content from MLR and CLR. The synthesized flavonoid from the plant matrix is transported to the vacuoles and cell walls as reported by Agati et al. [12]. However, the acidic environment could result in the disruption of the cellulosic cell walls by breaking the chemical bonds between the flavonoid and cell walls [13]. This subsequently led to an eventual exudation of the flavonoid content and their solubilization in the extraction solvent. 

Figure 2a,b shows the effect of pH variation on the in vitro DPPH radical scavenging activity and ferric-reducing antioxidant activities of the plant extracts. The result obtained from the China lotus root extract clearly depicted an inverse proportionality. The %DPPH_sc_ and FRAP value significantly dropped from 57.13 to 40.29%DPPH_sc_ and 4.745 to 3.086 mM FeSO_4_, respectively. Meanwhile, for MLR extract, a similar pattern was observed (61.86% to 38.50%DPPH_sc_ and 4.745 to 3.086 mM FeSO_4_, respectively). These results suggest that flavonoid extracted from the lotus roots is responsible for the free radical scavenging and ferric-reducing activity. Therefore, the high concentration of flavonoid extracted from the lotus roots is largely responsible for the strong antioxidant activities [14].

### 2.3. Effect of Extraction Time

In this study, extraction time was varied from 0.25 to 5 h, at constant pH, solvent-to-sample ratio and temperature. Extract from China lotus root gave the highest TFC at 1 h of extraction time (0.185 mg PCE/g sample), while the lowest TFC was obtained at 5 h of extraction time (0.124 mg PCE/g sample). On the other hand, extract from MLR gave the highest TFC at 0.5 h of extraction (0.237 mg PCE/g sample) while the lowest TFC was obtained at 5 h of extraction (0.114 mg PCE/g sample) (Figure 3).

The decrement suggested that the extension of extraction time could cause the released flavonoids to be exposed to the environment at high temperatures for a longer period, thus increasing their chance of decomposition. As discussed previously, antioxidant activities like DPPH radical scavenging and ferric-reducing activity are greatly influenced by the concentration of flavonoid extracted. Therefore, the trends of the two antioxidant activities conducted (Figure 4a,b) were observed to be similar to the trend obtained from TFC studies (Figure 3). From the result, extraction time within 1.5 h was chosen for optimization study. A similar result was obtained by Nepote et al. [15], who found that 10 min was the optimum time to extract phenolic compounds from peanut skin.

### 2.4. Effect of Solvent-to-Sample Ratio

A range of 10–50 mL/g solvent-to-sample ratio was studied in the extraction of phytochemicals from both Malaysia lotus (MLR) and China lotus (CLR) roots. In general, the increase in the solvent-to-sample ratio encouraged the increase in TFC and antioxidant activities in both Malaysia lotus (MLR) and China lotus (CLR) roots extracts. For China lotus root extract, the highest TFC was obtained with the solvent-to-sample ratio of 40 mL/g (0.393 mg PCE/g sample), followed by 50 mL/g (0.359 mg PCE/g sample) and 30 mL/g (0.207 mg PCE/g sample) (Figure 5).

On the other hand, the highest TFC content obtained from MLR extract was at a solvent-to-sample ratio of 50 mL/g (0.797 mg PCE/g sample), then 40 mL/g (0.699 mg PCE/g sample) and 30 mL/g (0.504 mg PCE/g sample). Moreover, the antioxidant activity of Malaysia lotus (MLR) and China lotus (CLR) roots increase as the solvent-to-sample increases (Figure 6). The result indicated that the increase in solvent volume is directly proportional to the antioxidant activities [4]. The result is consistent with the mass transfer principle, which describes that the concentration gradient between the sample and the solvent is the driving force during mass transfer [16]. Therefore, the higher buffer-to-sample ratio used will create a higher driving force for the extraction process. Thus, a larger volume of solvents will be able to extract more flavonoid compounds [17].

### 2.5. Effect of Temperature

TFC contents extracted from both Malaysia lotus (MLR) and China lotus (CLR) roots with their DPPH and FRAP assays are presented in Figure 7a–c. There were no significant differences in the range of extraction temperature at 25–60 °C. Thus, the effect of temperature was not studied in optimization studies and was set at room temperature as a constant in the further experiment.

### 2.6. Comparison of CLR and MLR Extracts

Malaysia lotus (MLR) and China lotus (CLR) roots were compared in this research as these two cultivars are the most common lotus root found in local markets in Malaysia. When comparing the physical appearance of raw lotus root, CLR showed a pinkish-beige and larger (in terms of diameter) root, whereas MLR is relatively slimmer and whitish-beige. However, when the root was cooked, the CLR turned into a purplish-brown colour, whereas MLR remained as beige colour and slightly translucent. The textures of both lotus roots were also different. Therefore, it was believed that these two cultivars of lotus root are different in compositions. According to the results, CLR extract shows higher TFC (0.699 mg PCE/g sample) compared to MLR extract (0.533 mg PCE/g sample) at pH 2. CLR extract also exhibited stronger free radical scavenging activity (21.96 %DPPH_sc_) compared to MLR extract (13.53 %DPPH_sc_) at low solvent-to-sample ratio (10 mL/g). At long extraction time (5 h), CLR extract also shows a more stable and higher reducing power (2.18 mM FeSO_4_) than MLR extract (1.73 mM FeSO_4_). These results could be due to the climate and botanical aspects of the plantation of different cultivars lotus root. According to the research conducted by Jang et al. [9], the growth of lotus in monsoon climate is slower than the growth in summer and thus affects the synthesis of flavonoids in the plant cell. Also, progressive changes in traits like the production of phytochemicals in the root could happen during seasonal variation especially from the dormant to the active phase. Many other factors could also influence the secondary metabolite profile of the root such as geographic location, age of the plant, season, associated microflora, nutritional status, and environmental stress [17]. Ultimately, the result obtained clearly revealed CLR as a better source of water-soluble flavonoids when compared to MLR. However, the optimization study was subsequently conducted to obtain the highest flavonoid content. 

### 2.7. Optimization Study

The effects of extraction factors, i.e., pH (X_1_), time (X_2_), and solvent-to-sample ratio (X_3_), were included in the optimization study. These parameters (X_1_: pH 2–3; X_2_: 0.5–1.5 h; X_3_: 20–40 mL/g) were selected during the preliminary studies (Section 3.1). The three responses of interest were TFC, %DPPH_sc_, and FRAP. The results of 17 runs using BBD are shown in Table 1a,b and this includes the design of the experiment, observed responses and predicted values. A close agreement between experimental and predicted values was subsequently attained. From the results, TFC ranged from 0 to 0.68 mg PCE/g CLR extract and 0.1–0.58 mg PCE/g MLR extract were reported. For CLR extract, the highest TFC value (0.68 mg PCE/g sample) was achieved under condition of X_1_ = pH 2.5, X_2_ = 1.5 h and X_3_ = 40 mL/g. Meanwhile, for MLR extract, the highest TFC value was of 0.58 mg PCE/g sample which was obtained under the condition of X_1_ = pH 2.5, X_2_ = 0.5 or 1.5 h and X_3_ = 40 mL/g. As for the antioxidant activities, the ranges of %DPPH_sc_ of CLR and MLR extracts were reported as 7.2–50.5 %DPPH_sc_ and 12.88–30.22 %DPPH_sc_, respectively. Both highest %DPPH_sc_ (50.5 and 30.22 %DPPH_sc_) from CLR and MLR extracts, respectively, were found in conditions of X_1_ = pH 2.5, X_2_ = 30 min and X_3_ = 40 mL/g. At the same condition, CLR extract gave the highest FRAP value (2.20 mM FeSO_4_), but MLR extract gave the highest FRAP value (1.96 mM FeSO_4_) at condition X_1_ = pH 2.5, X_2_ = 60 min and X_3_ = 30 mL/g. These conditions varied depending on the response required. The optimum extraction condition was therefore investigated in order to obtain desirable TFC, %DPPH_sc_, and FRAP value.

### 2.8. Model Fitting

Table 1a,b presents the results of quadratic models fitted to the data. The results of the analysis of variance (ANOVA) indicated that the contribution of the quadratic model was significant for the responses. Equations (1), (3), and (5) represent the fitted quadratic models for TFC, %DPPH_sc_, and FRAP of CLR and MLR extracts; on the other hand, Equations (2), (4), and (6) are of MLR extracts, respectively. The significance of each coefficient was determined using the F-test and *p*-value in Table 2a,b. If the absolute F-value becomes greater and the *p*-value becomes smaller, the corresponding variables would be more significant [4]. 

It could be observed that the largest effects on the TFC of CLR extract were the linear term of solvent-to-sample ratio ($x_{3}$) and quadratic term of pH ($x_{1}^{2}).$ The results shown in Table 2a suggested that the change of pH and solvent-to-sample had significant effects (*p* < 0.05) on the TFC of CLR extract. Time, on the other hand, did not have any significant (*p* > 0.05) contribution in TFC of CLR extract. The coefficient of determination (r^2^) of the predicted models in this response of CLR was 0.9765, indicating a good correlation of the data to the model. However, the *p*-value for lack of fit was 0.0264 with a lack of good fit for the mathematical models in Equation (1) 

(1)$$\begin{array}{l} TFC=\;0.59+0.034x_{1}+0.046x_{2}+0.091x_{3}-0.043x_{1}^{2}-0.055x_{2}^{2}+0.019x_{3}^{2}+8.96\times 10^{-3}x_{1}x_{2}\\ \;\;\;\;\;\;+0.057x_{1}x_{3}+2.29\times 10^{-3}x_{2}x_{3} \end{array}$$

However, the significant effects on the TFC of MLR extract have linear term for pH $\left(x_{1}\right)$, linear term for solvent-to-sample ratio $\left(x_{3}\right)$, quadratic term for pH $\left(x_{1}^{2}\right)$, quadratic term for solvent-to-sample ratio $\left(x_{3}^{2}\right)$, interaction term for pH and time $\left(x_{1}x_{2}\right)$ and interaction term for time and solvent-to-sample ratio $\left(x_{2}x_{3}\right)$ (Table 2b). The results showed that all three factors (pH, time and solvent-to-sample ratio) contributed significantly (*p* < 0.05) in TFC of MLR extract. The coefficient of determination (r^2^) of the predicted models in the response of MLR was 0.9873. Although the *p*-value for lack of fit (0.0003) shows not a good fit for mathematic models in Equation (2), verification of an optimized extraction conditions will be carried out.

(2)$$\begin{array}{l} TFC=\;0.49-0.059x_{1}-0.018x_{2}+0.11x_{3}-0.21x_{1}^{2}-5.17\times 10^{-3}x_{2}^{2}\\ \;\;\;\;\;\;\;-0.036x_{3}^{2}-0.036x_{1}x_{2}-0.024x_{1}x_{3}+0.03x_{2}x_{3} \end{array}$$

For CLR extract, it could be seen that the quadratic term of pH ($x_{1}^{2})$ showed the largest effect on %DPPH_sc_, followed by the linear term of solvent-to-sample ratio ($x_{3})$ and the interaction term of pH and solvent-to-sample ratio ($x_{1}x_{3})$. The coefficient of determination (r^2^) of the predicted models in the response was 0.9937. However, the *p*-value for the lack of fit was 0.0115, which suggested not a good fit for the mathematical model Equation (3).
(3)$$\begin{array}{l} \%DPPHsc=\;45.46-1.00x_{1}-0.51x_{2}+3.26x_{3}-31.34x_{1}^{2}-1.53x_{2}^{2}\\ \;\;\;\;\;\;\;+0.48x_{3}^{2}-2.18x_{1}x_{2}+3.73x_{1}x_{3}-0.016x_{2}x_{3} \end{array}$$

The linear term of pH ($x_{1})$ showed the most significant (*p* < 0.05) effect on %DPPH_sc_ of MLR extract, followed by a linear term of solvent-to-sample ratio ($x_{3})$, a quadratic term of pH ($x_{1}^{2})$ and linear term of time ($x_{2})$. The other quadratic terms ($x_{2}^{2},x_{3}^{2})$ and all the interaction terms $x_{1}x_{2},\;x_{1}x_{3}$ and $x_{2}x_{3}$ did not have significant (*p* > 0.05) effects. The coefficient of determination (r^2^) of the predicted models in this response was 0.9474, whereas the *p*-value for lack of fit was 0.1988, which suggests a relatively good fit to the mathematic models in Equation (4).
(4)$$\begin{array}{l} \%DPPHsc=\;25.34-4.84x_{1}-1.70x_{2}+4.08x_{3}-4.27x_{1}^{2}+0.37x_{2}^{2}-1.14x_{3}^{2}-1.40x_{1}x_{2}-0.076x_{1}x_{3}\\ \;\;\;\;\;\;\;-1.76x_{2}x_{3} \end{array}$$

In terms of FRAP assay, the only variable with significant (*p* < 0.05) effect on FRAP value of CLR extract was the quadratic term of pH ($x_{1}^{2})$, whereas other terms were not significant (*p* > 0.05). The coefficient of determination (r^2^) of the predicted model was 0.9212 and the *p*-value for lack of fit was 0.4533, which was a good fit for the mathematical model in Equation (5).
(5)$$\begin{array}{l} FRAP=\;1.87-0.011x_{1}-0.031x_{2}+0.11x_{3}-0.85x_{1}^{2}-0.034x_{2}^{2}+0.053x_{3}^{2}-0.073x_{1}x_{2}+0.14x_{1}x_{3}\\ \;\;\;\;\;\;-0.031x_{2}x_{3} \end{array}$$

Meanwhile, for MLR extract, the variables that gave the largest effect on the FRAP value was the quadratic term of pH ($x_{1}^{2})$, followed by a linear term of solvent-to-sample ratio ($x_{3})$ and linear term of pH ($x_{1})$. Time, on the other hand, did not contribute any significant (*p* > 0.05) effect to the response. The coefficient of determination (r^2^) was 0.8865 and the *p*-value for lack of fit was 0.9720, which suggested that it was an excellent fit for the mathematic model in Equation (6).
(6)$$\begin{array}{l} FRAP=\;1.71-0.13x_{1}-0.022x_{2}+0.18x_{3}-0.34x_{1}^{2}-3.96\times 10^{-3}x_{2}^{2}-0.055x_{3}^{2}-0.027x_{1}x_{2}-0.015x_{1}x_{3}\\ \;\;\;\;\;\;-0.012x_{2}x_{3} \end{array}$$

### 2.9. Response Surface Model and Contour Plot

3D response surfaces shown in Figure 8, Figure 9 and Figure 10 demonstrated the changes in TFC, %DPPH_sc_ and FRAP values as a function of pH and solvent-to-sample ratio, while extraction time was set at 3 different levels (i.e., 0.5, 1 and 1.5 h). The 3D response surfaces for TFC of CLR extract as a function of pH and solvent-to-sample ratio were given in Figure 8a(i–iii). The overall results showed that the extraction time had no significant (*p* > 0.05) effect on TFC value. Results also showed that a higher solvent-to-sample ratio would give higher TFC yield, where at pH 2.5 and the buffer-to-sample ratio of 40:1, TFC reported the highest for the 0.679 mg PCE/g sample. As mentioned previously, TFC yield increased as pH increased from pH 2 to pH 2.5, ranging from 0 to 0.679 mg PCE/g sample and then decreased to 0.098 mg PCE/g sample as pH increased from pH 2.5 to pH 3. This suggests that pH 2.5 (as described previously) is the optimum pH for the extraction of flavonoids.

For the TFC of MLR extract, the 3D surfaces as a function of pH and extraction time are given in Figure 8b(i–iii). Results showed that at low solvent-to-sample ratio (20:1, *v*/*w*), an increase of TFC could be observed by increasing pH up to pH 2.5. Further increment in pH to pH 3.0 led to a decrease in TFC. A similar trend could be observed at a solvent-to-sample ratio at 30:1 and 40:1 (*v*/*w*). Extension in the extraction time showed no significance (*p* > 0.05) at all three levels of the solvent-to-sample ratio. However, the interaction between pH and extraction time showed a significant (*p* < 0.05) effect in this experiment. At low solvent-to-sample ratio (20:1, *v*/*w*), TFC increased with increasing pH up to 2.25 and extraction time (~1 h) and then decreased at higher pH (~pH 2.75) with further extended extraction period. A similar trend was found at a solvent-to-sample ratio of 30:1 (*v*/*w*). On the other hand, at a high solvent-to-sample ratio (40:1, *v*/*w*), increasing pH and extraction time led to higher TFC (~0.589 PCE/g sample). Longer extraction time showed higher efficiency in the extraction of flavonoids as it provides sufficient time for the solutes to expose to the extraction solvents. Similar results were obtained by Xu et al. [18]. These results suggested a positive effect of increasing extraction time (less than 2 h) on the extraction of flavonoid from lotus roots.

Furthermore, the 3D surfaces for %DPPH_sc_ of CLR extract displayed in Figure 9a(i–iii) showed similar results obtained in TFC. Overall, the time had no significant (*p* > 0.05) effects, while pH exhibited a major effect on DPPH radical scavenging activity. At fixed extraction time and solvent-to-sample ratio, a maximum %DPPH_sc_ could be observed at region pH 2.5. Lower %DPPH_sc_ were obtained at both the lower and higher pH regions. Interaction between pH and solvent-to-sample ratio contributed a significant (*p* < 0.05) effect on %DPPH_sc_. At 0.5 h extraction time, given the increment in solvent-to-sample ratio (from 20 mL/g to 40 mL/g) at low pH (pH 2), only a slight increment in %DPPH_sc_ (~7.74 %DPPH_sc_) was observed. However, when the pH increased up to 2.5 and above, an apparent increment in %DPPH_sc_ could be observed with the increment of the solvent-to-sample ratio. 

For MLR, 3D surfaces of %DPPH_sc_ showed that there was no significant (*p* > 0.05) effect in the interaction between factors. Significant (*p* < 0.05) effect could only be observed in pH. At all three levels of extraction time and solvent-to-sample ratio, increment in pH from 2 to 2.5 did not show any significant (*p* > 0.05) difference in %DPPH_sc_ (~18.81–22.81%), however, further increment in pH up to pH 3, %DPPH_sc_ were significantly (*p* < 0.05) decreased to ~6.58%. This result was similar to the result obtained from CLR. In general, the results suggested that these flavonoids were favored at ex traction conditions of pH 2.5 which could give a high DPPH radical scavenging activity [19].

Also, the 3D response surfaces for FRAP of CLR extract as a function of pH and solvent-to-sample ratio are given in Figure 10a(i–iii). In general, pH affected the activity significantly (*p* < 0.05); from all three levels of extraction time, FRAP value reached maximum (~1.63–1.93 mM FeSO_4_) at pH~2.25, then decreased with further increment of pH. On the other hand, the solvent-to-sample ratio did not increase significantly (*p* > 0.05) at all three levels of extraction time. The result showed that the interaction between pH and solvent-to-sample ratio did not significantly (*p* < 0.05) affect the FRAP of CLR. For FRAP of MLR extract, the 3D response surfaces as a function of pH and solvent-to-sample ratio are given in Figure 10b(i–iii). At 0.5 h of extraction time, increment in solvent-to-sample ratio (from 20:1 to 40:1, *v*/*w*) significantly (*p* < 0.05) increased FRAP value from (~1.059–1.681 mM FeSO_4_). At the same extraction time, the increment of pH up to pH 2.25 significantly (*p* < 0.05) increased the FRAP value, from ~1.266 to ~1.473 mM FeSO_4_ at 20 mL/g solvent-to-sample ratio. However, continuous increment from pH ~2.5 led to a significant (*p* < 0.05) decrement in FRAP (<1.059 mM FeSO_4_). Similar to the result obtained from CLR, the interaction between pH and solvent-to-sample ratio did not have a significant (*p* > 0.05) effect on FRAP assay. 

### 2.10. Validation of Predictive Models

Based on the findings, a verification study was performed to evaluate the optimal operating conditions for the extraction with a high yield of antioxidant content (TFC) and high antioxidant activities (%DPPH_sc_ and FRAP) with consideration of efficiency, energy conservation and feasibility of the experiment. An optimal condition to achieve high TFC and high antioxidant activities were determined for both CLR and MLR. Therefore, from the model, the optimal conditions for TFC, %DPPH_sc_ and FRAP of CLR extract were pH 2.5, extraction time of 0.5 h, and the solvent-to-sample ratio of 40 mL/g. On the other hand, the optimal conditions for TFC, %DPPH_sc_ and FRAP of MLR extract were pH 2.4, extraction time of 0.5 h, and the solvent-to-sample ratio of 40 mL/g. These conditions gave TFC, %DPPH_sc_ and FRAP of 0.599 mg PCE/g sample, 48.36 %DPPH_sc_ and 2.07 mM FeSO_4_ for CLR extract and 0.549 mg PCE/g sample, 29.11 %DPPH_sc_ and 1.89 mM FeSO_4_ for MLR extract, respectively. Experimental values for TFC, %DPPH_sc_ and FRAP obtained were of the 0.66 mg PCE/g sample, 42.78 %DPPH_sc_ and 1.89 mM FeSO_4_ for CLR extract and 0.59 mg PCE/g sample, 31.72 %DPPH_sc_ and 1.94 mM FeSO_4_ for MLR extract, respectively. It could be observed that only small deviations were found between the predicted values and experimental values [20]. Thus, the models could be used to optimize the extraction of flavonoids from both CLR and MLR.

## 3. Materials and Methods

### 3.1. Materials

Fresh Chinese and Malaysian lotus root cultivars were purchased from the local market located in Jelutong, Penang, Malaysia. The sample was thoroughly washed with deionized distilled water, subsequently cut into small cubes and lyophilized to remove inherent moisture. The lyophilized samples were then ground into a powder prior to further experiments. All chemicals used in this study were analytical grade and purchased from Sigma-Aldrich (Selangor, Malaysia) and Fluka (Industriestrasse 25, Switzerland).

### 3.2. Design of Experiments

During the preliminary study, different solvents including methanol, ethanol, acetone, ethyl acetate, hexane, water or enzyme solution (cellulase or hemicellulase, 1% *w*/*v*) were used to obtain the most efficient extraction condition. The effect of pH (2–6), temperature (25–60 °C), extraction time (0.25–5 h) and buffer-to-sample ratio (10–50 mL/g) was further investigated using the single-factor experiment. This was achieved by varying one factor and keeping other factors constant using the middle point viz: extraction time (2 h), pH (4), extraction temperature (40 °C), and solvent-to-sample ratio (20 mL/g). The selected factors and levels from the single-factor experiments were subsequently used for the optimization study using the Box–Behnken design.

The parameters of extraction were optimized using response surface methodology (RSM). Box–Behnken design (BBD) was applied while pH (X_1_: 2–3), extraction time (X_2_: 0.5–1.5 h) and solvent-to-sample ratio (X_3_: 20–40 mL/g) were selected as independent variables based on the results of preliminary studies. The experimental design consisted of twelve factorial points and each center point has five replicates (Table 2). Each experiment was carried out in triplicate and the mean values of the observed response. Experiments were conducted randomly to reduce the effects of unexpected variability in the observed responses. 

The variables were coded accordingly to the equation:(7)$$x\;=\;\frac{\left(X_{i}-X_{o}\right)}{\Delta X}$$
where *x* was the coded value, *X_i_* was the corresponding actual value, *X_o_* was the actual value in the center of the domain, and Δ*X* was the increment of *X_i_* corresponding to a variation of 1 unit of *x*.

The mathematical model corresponding to the composite design was:(8)$$Y\;=\;\beta_{o\;}+\;\sum_{i\;=\;1}^{3}\beta_{i}X_{i}+\sum_{i\;=\;1}^{3}\beta_{ii}X_{i}^{2}+\sum_{i\;=\;1}^{2}\sum_{j\;=\;1\;+\;1}^{3}\beta_{ij}X_{i}X_{j}+\varepsilon$$where Y indicated the dependent variables (TFC, %DPPH_sc_, and FRAP), $\beta_{o}$ is the model constant, $\beta_{i}$, $\beta_{ii}$ and $\beta_{ij}$ were the model coefficients which represent linear, quadratic or interaction effects of the variables, whereas ε was the error. This analysis of experimental design data and calculation of predicted responses were carried out using Design-Expert software (version 6.0, Stat-Ease Inc., Minneapolis, MN, USA). In order to verify the validity of the statistical experimental design, additional confirmation experiments were conducted.

### 3.3. Statistical Analysis

Comparison of means was performed by one-way analysis of variance (ANOVA) followed by Duncan’s test. Statistical analysis (*p* < 0.05) was performed using Statistical Package for Social Science for Windows, version 20.0 (IBM SPSS, Chicago, IL, USA). The optimal extraction conditions were estimated through regression analysis and three-dimensional response surface plots of the independent variables and each dependent variable.

### 3.4. Extract Preparation 

Briefly, 1 g of the plant sample was added into the extracting solvent at varied solvent-to-sample ratio and pH. The mixture was then homogenized and incubated at different temperatures for selected extraction time and constantly agitated at 250 rpm. After the extraction process had been completed, the mixture was then centrifuged at 4500 rpm for 15 min at 4 °C. The supernatant was then collected and reconstituted to 50 mL of the solvent. The crude extract was stored at −20 °C prior to future analysis. All samples were prepared in triplicate.

### 3.5. Determination of Total Flavonoid Content (TFC)

Total flavonoid content was determined by colorimetric assay modified from Kim et al. [21]. Distilled water (80 µL) was added to 20 µL of extract. Then, 5% (*w*/*v*) of sodium nitrite solution (6 µL) and 10% (*w*/*v*) aluminum chloride solution (6 µL) were added to the mixture. Mixtures were incubated at room temperature for 5 min. Then, 40 µL of 1 M NaOH was added to the mixture. The final volume of the reaction mixture was topped up to 200 µL with distilled water and the absorbance measured at 510 nm. All samples were prepared in triplicate. A standard curve was generated with pyrocatechol and results were expressed in mg pyrocatechol equivalent (PCE)/g sample.

### 3.6. DPPH Radical Scavenging Activity Determination

The scavenging activity on DPPH radical of rambutan peel extracts was determined by the modified methods of Liu et al. [22]. Each extract (5 µL) was added to 150 µL of DPPH solution (0.1 mM). The mixtures were incubated for 30 min in the dark at 30 °C and discolorations were determined at 517 nm. %DPPH scavenged (%DPPH_sc_) was calculated using Equation (9) below:(9)$$DPPH_{sc}=\frac{Abs_{control}-Abs_{sample}}{Abs_{control}}\;\times \;100$$
where *Abs_control_* was defined as absorbance of the control at 517 nm and *Abs_sample_* was defined as the absorbance value of the extracts at 517 nm.

### 3.7. Ferric-Reducing Power Assay (FRAP)

FRAP assay was conducted by referring to the method of Benzie and Strain [23]. The FRAP reagent was a mixture of 1 mL of 10 mM tripyridyltriazine (TPTZ) solution, 1 mL of 20 mM FeCl_3_·6H_2_O and 10 mL of 0.3 M acetate buffer at pH 3.6. FRAP reagent (200 µL) was pre-warmed at 37 °C and then mixed with 2.7 µL of the extract. The reaction mixtures were incubated at 37 °C for 1 h and absorbance at 593 nm was measured. All samples were prepared in triplicate. A standard curve using FeSO_4_ was obtained.

## 4. Conclusions

RSM was used to determine the optimum extraction parameters that would give the high yield of TFC and antioxidant activities. According to ANOVA, pH, extraction time and solvent-to-sample ratio showed different effects on the targeted responses depending on the types of lotus root used. Interactions of the extraction parameters were also found. Quadratic models were successfully developed and used in predicting all the responses. The optimal conditions based on the combinations of all responses were obtained (CLR: X_1_ = 2.5; X_2_ = 0.5 h; X_3_ = 40 mL/g; MLR: X_1_ = 2.4; X_2_ = 0.5 h; X_3_ = 40 mL/g). Results showed that the predicted and experimental values were similar. Therefore, it could be concluded that the models obtained could be used to optimize the extraction process of water-soluble flavonoids from both China lotus and Malaysian lotus root.

## Figures and Tables

**Figure 1 molecules-26-02014-f001:**
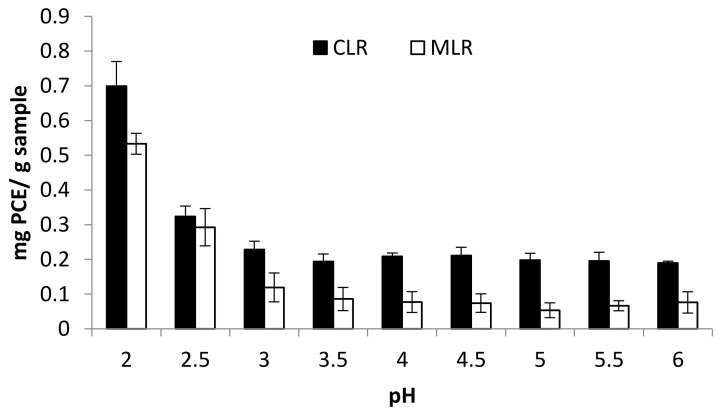
Effect of pH on total flavonoid content (TFC).

**Figure 2 molecules-26-02014-f002:**
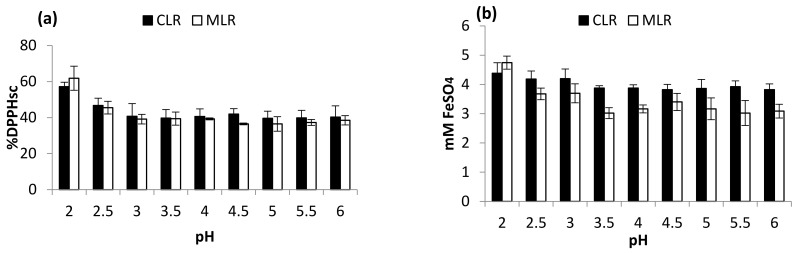
Effect of pH (**a**) DPPH scavenging activity and (**b**) ferric-reducing antioxidant power (FRAP).

**Figure 3 molecules-26-02014-f003:**
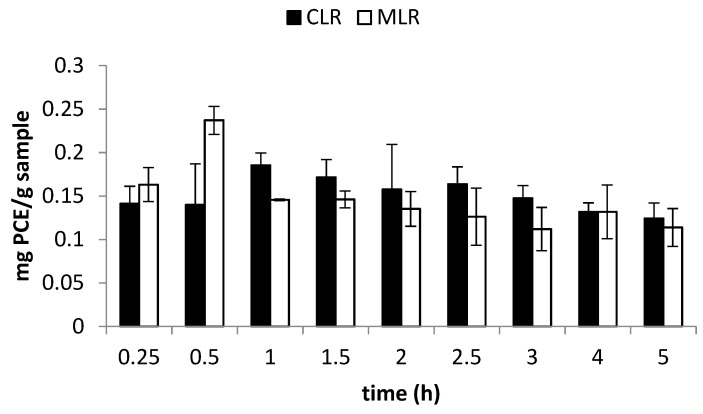
Effect of extraction time on total flavonoid content (TFC).

**Figure 4 molecules-26-02014-f004:**
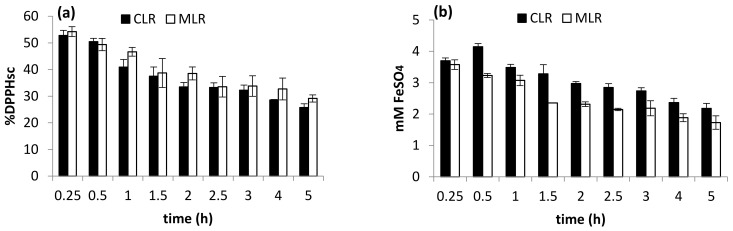
Effect of extraction time on (**a**) DPPH scavenging activity and (**b**) ferric-reducing antioxidant power (FRAP).

**Figure 5 molecules-26-02014-f005:**
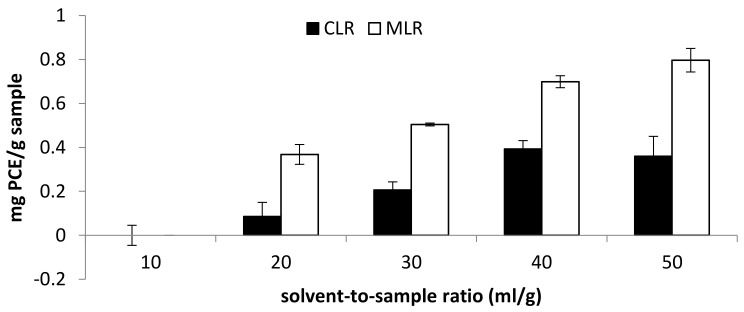
Effect of solvent-to-sample ratio on total flavonoid content (TFC).

**Figure 6 molecules-26-02014-f006:**
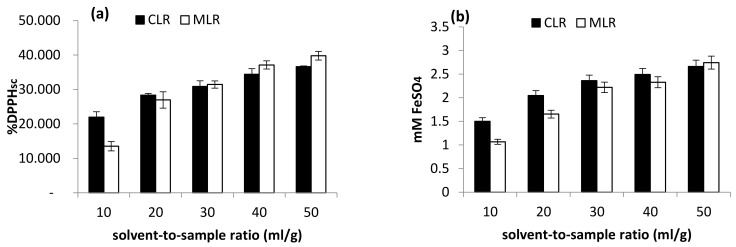
Effect of solvent-to-sample ratio on (**a**) DPPH scavenging activity and (**b**) ferric-reducing antioxidant power (FRAP).

**Figure 7 molecules-26-02014-f007:**
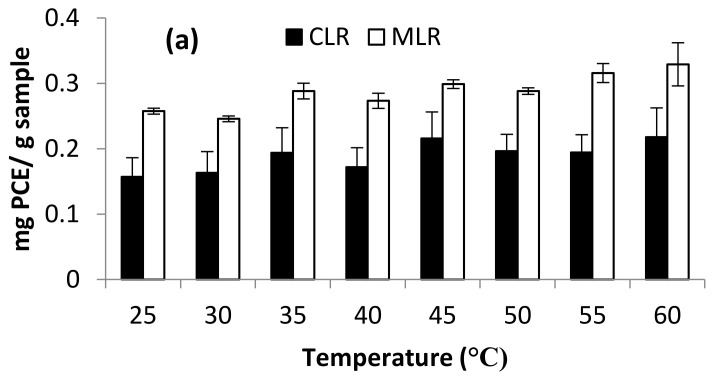
Effect of temperature on (**a**) Total flavonoid content (**b**) DPPH scavenging activity and (**c**) ferric-reducing antioxidant power (FRAP).

**Figure 8 molecules-26-02014-f008:**
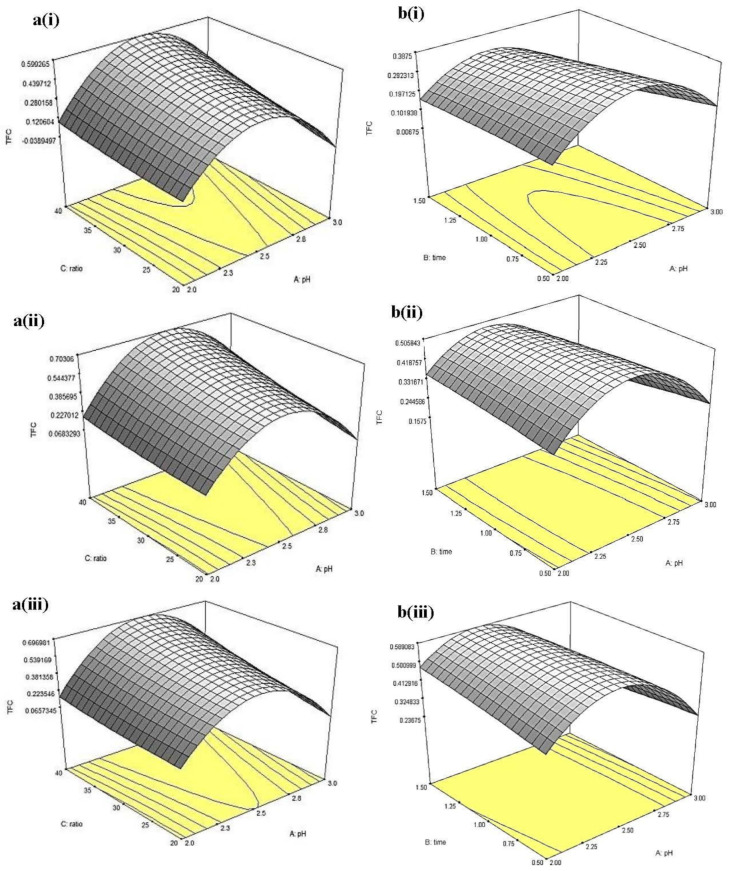
Three-dimensional response surfaces for TFC of (**a**) China cultivar of lotus root at (i) 0.5, (ii) 1, (iii) 1.5 h extraction time as a function of pH and solvent-to-sample ratio and (**b**) Malaysia cultivar of lotus root at (i) 20, (ii) 30, and (iii) 40 (mL/g) of solvent-to-sample ratio as a function of pH and extraction time.

**Figure 9 molecules-26-02014-f009:**
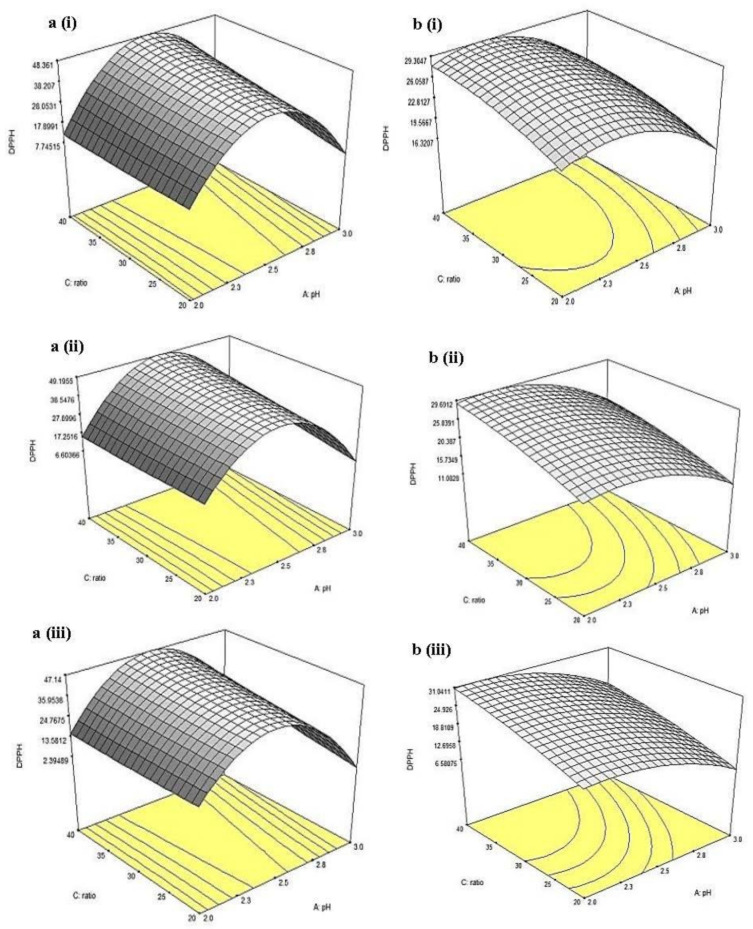
Three-dimensional response surfaces for %DPPH_sc_ of (**a**) China cultivar of lotus root at (i) 0.5, (ii) 1, (iii) 1.5 h extraction time and (**b**) Malaysia cultivar of lotus root at (i) 0.5, (ii) 1, and (iii) 1.5 h extraction time as a function of pH and solvent-to-sample ratio.

**Figure 10 molecules-26-02014-f010:**
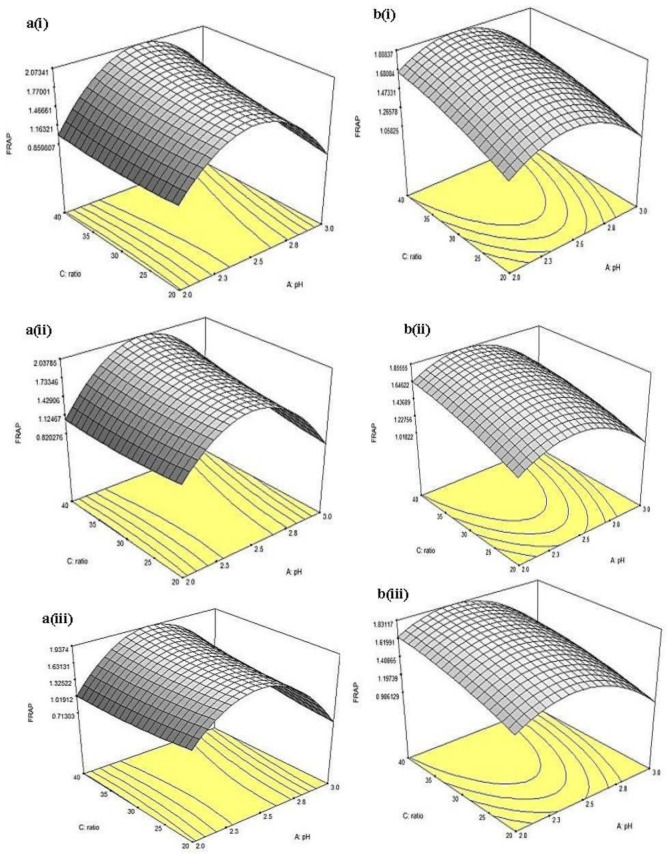
Three-dimensional response surfaces for FRAP of (**a**) China cultivar of lotus root at (i) 0.5, (ii) 1, (iii) 1.5 h extraction time and (**b**) Malaysia cultivar of lotus root at (i) 0.5, (ii) 1, and (iii) 1.5 h extraction time as a function of pH and solvent-to-sample ratio.

**Table 1 molecules-26-02014-t001:** ANOVA for response surface models: estimated regression model of relationship between response variables (TFC, %DPPH_sc_ and FRAP) and independent variables (X_1_,X_2_,X_3_) from (a) China lotus root and (b) Malaysia lotus root.

**(a) China Lotus Root**					
**Source**	**Sum of Squares**	**DF**	**Mean Square**	**F-Value**	***p*-Value**
TFC (mg PCE/g)a					
Model	0.89	9	0.10	32.32	<0.0001
Quadratic	0.79	3	0.26	85.51	<0.0001
A	0.01	1	0.01	3.03	0.1254
B	0.02	1	0.02	5.44	0.0525
C	0.07	1	0.07	21.53	0.0024
A2	0.76	1	0.76	248.23	<0.0001
B2	0.01	1	0.01	4.14	0.0814
C2	0.00	1	0.00	0.51	0.5001
AB	0.00	1	0.00	0.10	0.7558
AC	0.01	1	0.01	4.20	0.0796
BC	0.00	1	0.00	0.01	0.9364
Residual	0.02	7	0.00		
Lack of Fit	0.02	3	0.01	9.67	0.0264
Total	0.91	16			
%DPPHscb					
Model	4355.20	9	483.91	121.87	<0.0001
Quadratic	4185.33	3	1395.11	351.35	<0.0001
A	8.07	1	8.07	2.03	0.1969
B	2.05	1	2.05	0.52	0.4957
C	85.12	1	85.12	21.44	0.0024
A2	4134.86	1	4134.86	1041.35	<0.0001
B2	9.90	1	9.90	2.49	0.1583
C2	0.96	1	0.96	0.24	0.6374
AB	19.09	1	19.09	4.81	0.0644
AC	55.53	1	55.53	13.99	0.0073
BC	0.00	1	0.00	0.00	0.9879
Residual	27.79	7	3.97		
Lack of Fit	25.59	3	8.53	15.50	0.0115
Total	4383.00	16			
FRAP (mM)c					
Model	3.26	9	0.36	9.09	0.0041
Quadratic	3.05	3	1.02	25.55	0.0004
A	0.00	1	0.00	0.02	0.8793
B	0.01	1	0.01	0.19	0.6725
C	0.09	1	0.09	2.29	0.1736
A2	3.02	1	3.02	75.99	<0.0001
B2	0.00	1	0.00	0.12	0.7379
C2	0.01	1	0.01	0.30	0.6010
AB	0.02	1	0.02	0.54	0.4865
AC	0.08	1	0.08	1.99	0.2010
BC	0.00	1	0.00	0.10	0.7651
Residual	0.28	7	0.04		
Lack of Fit	0.12	3	0.04	1.08	0.4533
Total	3.53	16			
**(b) Malaysia Lotus Root**					
**Source**	**Sum of Squares**	**DF**	**Mean Square**	**F-Value**	***p*-Value**
TFC (mg PCE/g)d					
Model	0.35	9	0.038	55.79	<0.0001
Quadratic	0.21	3	0.069	100.21	<0.0001
A	0.03	1	0.029	41.83	0.0003
B	0.00	1	0.00	3.56	0.1012
C	0.10	1	0.097	140.58	<0.0001
A2	0.19	1	0.20	284.64	<0.0001
B2	0.00	1	0.00	0.20	0.6666
C2	0.01	1	0.00	7.82	0.0267
AB	0.01	1	0.00	8.17	0.0244
AC	0.00	1	0.00	2.94	0.1301
BC	0.00	1	0.00	4.39	0.0743
Residual	0.00	7	0.00		
Lack of Fit	0.00	3	0.00	4.40	0.0930
Total	0.35	16			
%DPPHsce					
Model	448.98	9	49.89	14.02	0.0011
Quadratic	84.57	3	28.19	7.92	0.0119
A	187.89	1	187.89	52.79	0.0002
B	23.15	1	23.15	6.51	0.0381
C	133.01	1	133.01	37.37	0.0005
A2	76.69	1	76.69	21.55	0.0024
B2	0.57	1	0.57	0.16	0.7015
C2	5.47	1	5.47	1.54	0.2549
AB	7.87	1	7.87	2.21	0.1807
AC	0.022	1	0.022	0.00	0.9389
BC	12.46	1	12.46	3.50	0.1035
Residual	24.91	7	3.56		
Lack of Fit	16.22	3	5.41	2.49	0.1988
Total	473.83	16			
FRAP (mM)f					
Model	0.92	9	0.10	6.12	0.0130
Quadratic	0.51	3	0.17	10.12	0.0061
A	0.14	1	0.14	8.42	0.0229
B	0.00	1	0.00	0.22	0.6559
C	0.26	1	0.26	15.75	0.0054
A2	0.48	1	0.48	28.91	0.0010
B2	0.00	1	0.00	0.00	0.9511
C2	0.013	1	0.013	0.79	0.4032
AB	0.00	1	0.00	0.18	0.6831
AC	0.00	1	0.00	0.073	0.7942
BC	0.00	1	0.00	0.054	0.8230
Residual	0.12	7	0.017		
Lack of Fit	0.01	3	0.00	0.071	0.9723
Total	1.04	16			

**Table 2 molecules-26-02014-t002:** BBD with the observed responses and predicted values for TFC (mg PCE/g), %DPPH_sc_ and FRAP (mM) for (a) China lotus root and (b) Malaysia lotus root.

Run	Variable Levels			(a) China Lotus Root						(b) Malaysia Lotus Root					
				Observed (*Y*_1_) ^a^			Predicted (Y_0_)			Observed (*Y*_1_) ^a^			Predicted (Y_0_)		
	X_1_ (pH)	X_2_ (time)	X_3_ (ratio)	TFC (mg PCE/g)	%DPPH_sc_	FRAP (mM)	TFC (mg PCE/g)	%DPPH_sc_	FRAP (mM)	TFC (mg PCE/g)	%DPPH_sc_	FRAP (mM)	TFC (mg PCE/g)	%DPPH_sc_	FRAP (mM)
1	2.5	0.5	40	0.66	50.5	2.20	0.60	48.18	2.06	0.58	30.22	1.84	0.55	28.58	1.88
2	2.5	0.5	20	0.43	41.4	1.89	0.42	41.63	1.79	0.38	23.54	1.48	0.39	23.95	1.49
3	2.5	1	30	0.62	45.7	2.10	0.59	45.46	1.87	0.52	26.41	1.72	0.49	25.34	1.71
4	3	1	40	0.34	18.6	1.31	0.36	20.58	1.32	0.26	18.83	1.36	0.27	19.09	1.35
5	3	1	20	0.10	7.2	0.86	0.07	6.60	0.82	0.12	12.88	1.00	0.1	11.08	1.02
6	2	0.5	30	0	10.2	0.86	0.03	11.91	0.96	0.31	26.74	1.52	0.31	26.58	1.50
7	2	1.5	30	0.15	15.6	1.18	0.11	15.27	1.05	0.37	27.37	1.49	0.35	25.98	1.51
8	2.5	1	30	0.55	45.2	1.88	0.59	45.46	1.87	0.48	25.67	1.54	0.49	25.34	1.71
9	3	1.5	30	0.23	10.6	0.98	0.20	8.89	0.88	0.16	13.33	1.18	0.16	13.49	1.20
10	2.5	1.5	20	0.44	38.3	1.64	0.50	40.65	1.79	0.27	15.38	1.51	0.3	17.02	1.47
11	2.5	1	30	0.58	44.4	1.72	0.59	45.46	1.87	0.49	25.70	1.76	0.49	25.34	1.71
12	2	1	20	0.14	18.1	1.13	0.11	16.06	1.12	0.18	20.88	1.24	0.17	20.62	1.25
13	2.5	1	30	0.61	45.5	2.03	0.59	45.46	1.87	0.5	26.15	1.96	0.49	25.34	1.71
14	2	1	40	0.15	14.6	1.02	0.18	15.14	1.06	0.41	27.13	1.67	0.44	28.93	1.65
15	2.5	1	30	0.58	46.5	1.64	0.59	45.46	1.87	0.48	22.76	1.58	0.49	25.34	1.71
16	2.5	1.5	40	0.68	47.4	1.84	0.69	47.14	1.94	0.58	29.12	1.81	0.57	28.71	1.81
17	3	0.5	30	0.04	13.9	0.95	0.08	14.27	1.09	0.25	18.31	1.32	0.27	19.70	1.29

^a^ Mean of triplicate determination.

## Data Availability

Not applicable.

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
