# Peer review of "Statistical Optimization of Flavonoid and Antioxidant Recovery from Macerated Chinese and Malaysian Lotus Root (Nelumbo nucifera) Using Response Surface Methodology"

_molecules, 2021, doi:10.3390/molecules26072014_

Round 1

Reviewer 1 Report

Statistical Optimization of Flavonoid and Antioxidant Recovery from Macerated Chinese and Malaysian Lotus Root (Nelumbo nucifera) using Response Surface Methodology

I would like to say that I appreciate the enormous effort realized by the authors of the manuscript. Nowadays, many people are aware of the benefits the natural antioxidants present. Therefore, many food and nutraceutical industries are interest in obtaining antioxidant compounds as new ingredients for their products. A lot of high plants are rich in flavonoids. This study employed Chinese and Malaysian lotus root (Nelumbo nucifera) as a natural matrix for extracting flavonoids. The main goal of this work was to determinate the optimum extraction parameters to obtain the higher yield of flavonoid contents and antioxidant activities. Response Surface Methodology (RSM) was employed to determine those parameters. Finally, according to results obtained, authors suggest the models obtained could be used to optimize the extraction process of flavonoids from CLR and MLR.

Introduction

I like the introduction, it is short, well-structured and concise. Regarding the content, the introduction is correct.

Materials and methods

In my opinion, this section is correct. The methodology is clear.

Results and discussion

I like the structure employed in this section, and the sub-sections used. As regards the results and discussion, this section is correct.

Conclusions

This section is correct; it summarizes well the content of the manuscript.

Therefore, the manuscript, generally, is well written and with excellent information. However, I provide some suggestions in order to enhance the quality of manuscript:

- Line 53: remove comma.

- Line 72: “the local lotus root grown in Malaysia was never been explored”. Incorrect grammatical: remove “been”.

- Line 75: “MLR and CLR”. It is necessary to add the meaning of these acronyms.

- Line 131: “centrifuged at 4500 g for 15 min…” Incorrect units Change “g” by “rpm”

- Line 283: “the design of bthe experiment”. Incorrect grammatical: change “bthe” by “the”

- Line 524: “predict values and predicted values”. According to manuscript context, it is correct “predict values and experimental values” or “predict and experimental values”.

Regarding the figures 1-7, I would advise changing the use the features to distinguish between CLR and MLR, for example, I would use black for the bar corresponding to CLR and white for MLR. The goal is to make easy for the reader to distinguish between CLR and MLR.

Regarding the figures 8, 9 and 10, it is necessary to improve the quality of the figures. For example, increase the word size of the axis.

Final remarks

Generally, this manuscript is well written and without big mistakes of spelling signs or in the way of writing. However, I suggest that the manuscript be revised according to the previous suggestions to improve the document.

Therefore, in my honest opinion, I consider that manuscript needs MINOR revisions before publication.

Author Response

Dear Editor and Reviewers,

Thank you for your useful comments and suggestions on our manuscript. We have modified (in red font/purple) the manuscript accordingly, and detailed corrections are listed below point by point (highlighted in yellow):

Reviewer #1

Therefore, the manuscript, generally, is well written and with excellent information. However, I provide some suggestions in order to enhance the quality of the manuscript:

  1. Line 53: remove the comma.
  • Amended accordingly as: global deaths in the year 2008 alone as reported by Cancer Research UK [1].
  1. Line 72: “the local lotus root grown in Malaysia was never been explored”. Incorrect grammatical: remove “been”.
  • Amended accordingly as: “It is important to know that the lotus species grown in Malaysia was never explored.”
  1. Line 75: “MLR and CLR”. It is necessary to add the meaning of these acronyms.
  • Amended accorsingly as: “…Malaysian lotus roots (MLR) and the Chinese lotus roots (CLR)….”
  1. Line 131: “centrifuged at 4500 g for 15 min…” Incorrect units Change “g” by “rpm”
  • Amended as: “…centrifuged at 4500 rpm…”
  1. Line 283: “the design of bthe experiment”. Incorrect grammatical: change “bthe” by “the”
  • Amended as “…design of the experiment…”
  1. Line 524: “predict values and predicted values”. According to manuscript context, it is correct “predict values and experimental values” or “predict and experimental values”.
  • Amended as “…between the predicted values and experimental values…”
  1. Regarding the figures 1-7, I would advise changing the use the features to distinguish between CLR and MLR, for example, I would use black for the bar corresponding to CLR and white for MLR. The goal is to make easy for the reader to distinguish between CLR and MLR.
  • All the figure legend (Figure 1-7) has been changed to black and white for CLR and MLR, respectively.
  1. Regarding the figures 8, 9 and 10, it is necessary to improve the quality of the figures. For example, increase the word size of the axis.
  • The resolution of the images (Figure 8-10) has been sharpened and improved.

 General response: The manuscript has been thoroughly amended in accordance with the suggestion from the reviewers. Other important information has been added to further enrich the manuscript. We look forward to your positive response.

Yours Sincerely

Assoc. Prof. Dr. Gan Chee-Yuen

Analytical Biochemistry Research Centre (ABrC), Universiti Sains Malaysia (USM), 11800 Gelugor,

Penang, Malaysia. Email: cygan@usm.my

Reviewer 2 Report

From a scientific point of view, the idea and intent to optimize the extraction process of these components is interesting. The article is well written, although sometimes the excess statistical data and acronyms make it a little confusing, especially in the results and discussión section. In this way, and since the results are consistent and well justified, the greatest criticism I can make to the manuscript and where I believe it should be improved is in its presentation.

  • Several acronyms that are not defined are included in the abstract. Please define them before using the abbreviation
  • All figure captions must be self-explanatory, showing all acronyms and symbols
  • All figures must be in colors or be more clearer
  • Please replace "hr" by "h" in all the manuscript

I would also like to know why, if the raw materials differ in composition, they have not been analyzed.

Author Response

Reviewer #2

From a scientific point of view, the idea and intent to optimize the extraction process of these components is interesting. The article is well written, although sometimes the excess statistical data and acronyms make it a little confusing, especially in the results and discussion section. In this way and since the results are consistent and well justified, the greatest criticism I can make to the manuscript and where I believe it should be improved is in its presentation.

  1. Several acronyms that are not defined are included in the abstract. Please define them before using the abbreviation
  • All acronyms have been defined accordingly
  1. All figure captions must be self-explanatory, showing all acronyms and symbols
  • Amended accordingly
  1. All figures must be in colors or be more clearer
  • The resolution of the images (Figure 8-10) has been sharpened and improved.
  1. Please replace "hr" by "h" in all the manuscript
  • Amended accordingly

  1. I would also like to know why, if the raw materials differ in composition, they have not been analysed.
  • The physical appearance of raw Chinese lotus root showed a pinkish-beige and larger (in terms of diameter) root whereas Malaysian lotus root is relatively slimmer and with whitish-beige. However, when the root was cooked, the CLR turned into a purplish-brown colour whereas MLR remained as beige colour and slightly translucent. The textures of both lotus roots were also different. Therefore, it was believed that these two cultivars of lotus root are different in compositions.
  • Moreover, this study only investigated the optimal parameter combination required to recover the total flavonoid and antioxidant content in Chinese and Malaysian lotus roots. This allowed us to make an informed comparison in determining the species with the potential of higher medicinal value.

 General response: The manuscript has been thoroughly amended in accordance with the suggestion from the reviewers. Other important information has been added to further enrich the manuscript. We look forward to your positive response.

Yours Sincerely

Assoc. Prof. Dr. Gan Chee-Yuen

…………………………………

Analytical Biochemistry Research Centre (ABrC),

Universiti Sains Malaysia (USM), 11800 Gelugor,

Penang, Malaysia

Email: cygan@usm.my

Round 2

Reviewer 2 Report

In my opinion, the authors have improved the manuscript taking into account the suggestions and questions of the reviewers. For this reason, I believe that he should be considered for publication on Molecules